# Cognitive Functions, Theory of Mind Abilities, and Personality Dispositions as Potential Predictors of the Detection of Reciprocity in Deceptive and Cooperative Contexts through Different Age Groups

**DOI:** 10.3390/bs13121007

**Published:** 2023-12-10

**Authors:** Anne-Lise Florkin, Alessia Rosi, Serena Lecce, Elena Cavallini

**Affiliations:** Department of Brain and Behavioral Sciences, University of Pavia, 27100 Pavia, Italy; anne-lise.florkin01@universitadipavia.it (A.-L.F.); serena.lecce@unipv.it (S.L.); elena.cavallini@unipv.it (E.C.)

**Keywords:** aging, personality, reciprocity, Theory of Mind

## Abstract

Reciprocity is a fundamental element in social interactions and implies an adequate response to the previous actions of our interactant. It is thus crucial to detect if a person is cooperating, deceiving, or cheating, to properly respond. However, older adults have been shown to have a lower ability to detect reciprocity compared to younger adults, partially tying this decline to cognitive functions. Another likely association to reciprocity in literature is made with personality dispositions, i.e., agreeableness, altruism, and empathic concern, and Theory of Mind (ToM). Consequently, the present study investigated age-related differences in the detection of the different components of reciprocity, as well as examined the predictors of reciprocity, such as cognitive measures, personality dispositions, and true and false beliefs in young (*n* = 98; 20–39 years), middle-aged (*n* = 106; 40–64 years), and older adults (*n* = 103; 65–96 years). The Mind Picture Story-Theory of Mind Questionnaire was used to measure the reciprocity components and true and false beliefs in each group. This study reported a significant decline in reciprocity detection from adults aged 65 years old and over. Additionally, the ability to detect reciprocity was significantly linked to cognitive functioning and ToM across all age groups, especially in older adults.

## 1. Introduction

Reciprocity is a fundamental component of social functioning. It can be described by a continuance of shared and appropriate social behaviors between interacting individuals [1,2]. It can take multiple forms and be direct, indirect, or generalized. Direct reciprocity is based on the ongoing interaction between two protagonists [3]. On the other hand, indirect reciprocity is based on reputation, it is a continuous evaluation of each other’s behavior by all the actors who contribute to maintaining the collective system [3,4,5]. Alternatively, generalized reciprocity is about the overall social interactions an individual had in the past, it is a chain of reciprocal behavior where a person reciprocates behaviors of previous interaction partners to the following interaction partners [4,6]. Importantly, reciprocity includes an expectation of a symmetrical response to an event, a sort of tit-for-tat. This conditional nature of reciprocity, relying on the valence of the behavior being mirrored [7], marks the significance of detecting cooperation, as well as deception and cheating.

Detecting this distinction becomes particularly crucial for older adults. In fact, cooperation is perceived by older adults as essential for maintaining participation and connectedness within their community [8]. On the other hand, the lack of detection of deception and cheating makes older adults more vulnerable to social tricks [9]. However, limited studies involving older adults investigated reciprocity or its underlying components. Moreover, research on reciprocity in older adults employed a combined measure of Theory of Mind (ToM) and reciprocity, namely, a ToM picture sequencing task [10,11], to the extent of considering reciprocity and ToM as a single element. Notably, Calso et al. [11,12] observed a decline in reciprocity, integrating true and false beliefs in the measurement, in a deceptive and cooperative context. In addition, Raimo et al. [13] used reciprocity as a general measure of cognitive ToM, aligning their findings with the previous observations of Calso et al. [11,12], indicating a decline in older adults in reciprocity as a subscale of a ToM measure. While these components have been previously explored in older adults in terms of both ToM and reciprocity, the present study takes a distinct approach. We use the same task to measure age differences in reciprocity, dividing the pure components of reciprocity from ToM. This differentiation would contribute to the investigation of reciprocity in aging, given the scarcity of research, and facilitate the examination of the association between reciprocity, as an aspect of social functioning, and one of its predictors, namely, ToM.

Indeed, reciprocal behavior requires people to understand and reason about others’ intentions and beliefs, which is defined as ToM [14]. An individual’s decision to reciprocate appropriately hinges on their beliefs about the other’s behavior, and consequently, their ability to infer mental states to the interacting partner [15]. More specifically, it is important to distinguish true from false beliefs. A cooperative person will demonstrate coherence between his intentions and actions which will generate a true belief, while a deceptive person hides the real intentions of their actions by generating a false belief about their actions. Therefore, being able to detect deception or cooperation depends on your ability to reason on beliefs. Nonetheless, it is important to acknowledge that other factors, such as cognition, can play a significant role in reciprocity detection.

In contrast to the two previously cited articles [12,13], centered on ToM and its link to cognition, our second focus in this study is to look at the possible role of ToM and other variables, such as cognition and personality traits, in predicting reciprocity. While understanding true and false beliefs may be associated with reciprocity, it has been demonstrated that reciprocal abilities are reliant on cognitive functions. Indeed, reciprocity is cognitively demanding since it requires remembering the reciprocal behavior of others and maintaining a balance in behaviors [5,16]. Working memory and fluid intelligence, measured through reasoning, seem to be two critical abilities involved in such behavior [17]. Studies have evidenced a correlation between cooperative behavior and an elevated fluid intelligence [18]. Moreover, an article on fraud victims [19] revealed that older adults’ decline in cognitive abilities, specifically working memory, play a key role in their difficulty to detect deception. 

Additionally, as reciprocity involves the interaction between at least two people, dispositional factors, such as personality traits, altruism, and empathic concern, could explain reciprocal behavior. Regarding traits, one of the most relevant ones involved in social interaction is agreeableness, part of the Big Five personality construct. It combines the dimensions of cooperation, kindness, politeness, and friendliness. Importantly, previous studies found a correlation between agreeableness and reciprocity [20,21,22]. For example, Sabater-Grande et al. [22] in their economic study on the Trust Game, found that a higher propensity to agreeableness is associated with a higher probability for reciprocal behavior. Moreover, Perugini et al. [20] found that the disposition of altruism was associated positively with positive reciprocity and negatively with negative reciprocity. However, the study on fraud victims revealed that older adults’ cognitive abilities significantly influenced their sensitivity towards scams, leading to a difficulty to detect deception, surpassing the role of personality dispositions [19].

### The Present Study

The main aim is the investigation of age-related differences in several critical aspects of reciprocity, namely, cooperation, deception, and cheating. The present study employs a highly relevant task, the Mind Picture Story-Theory of Mind Questionnaire (MPS-TOMQ) [10]. This task uniquely assesses all three aspects of reciprocity, along with true and false beliefs. Importantly, this instrument measures the individual’s ability to detect reciprocity, setting it apart from other tasks by only quantifying beliefs and perceptions [23,24]. Participants are asked to detect cooperation, deception, cheating, as well as true and false beliefs in three different stories: deception, cooperation, and a mix of both. This allows us to pinpoint the detection of each aspect of reciprocity within the various stories, facilitated by the specific coding of the task. Interestingly, reciprocity and ToM can be coded independently in the three different stories. 

Despite the relevance of investigating reciprocity in aging, few studies have analyzed it by comparing different age groups. Therefore, this cross-sectional study includes three age groups, younger, middle-aged, and older adults, to identify the starting point of the decline in the detection of cooperation, deception, and cheating.

The second aim of the present study is to investigate, whether ToM, cognitive abilities, and personality dispositions predict reciprocity performance and if these associations are different in the three age groups as a function of age. To achieve this, we used the true and false belief questions of the MPS-TOMQ as a measure of ToM, cognitive tests tapping fluid intelligence and working memory, and a series of personality questionnaires measuring individual differences in agreeableness, altruism, and emphatic concern. We first conduct a comparison analysis between the three age groups, and then run regression analyses to evaluate the predictive role of the identified factors within each age group separately.

As the literature reports a decline of older adults towards reciprocity [11,12,13,23,24], we expect them to have a lower performance in all the reciprocity components compared to middle-aged and younger adults. We also expect that the distinct types of stories in which the reciprocity dimensions can be detected will show divergent patterns of results due to the specific cognitive abilities linked to them [25]. For example, older adults will have difficulties in mixed stories because they are more cognitively demanding since they combine all the dimensions of reciprocity in one story. 

Concerning the predictive role of ToM, we expect true belief understanding to predict cooperation detection, while false belief will be a predictor of deception detection due to the nature of the concepts highlighted previously. Since ToM is a socio-cognitive concept and reciprocity is a social functioning concept, we expect that true and false beliefs will be significant predictors of the reciprocity dimensions. We suppose that this association is stronger in older adults for whom a decline in socio-cognitive abilities is linked to a decrease in social functioning [26].

As reciprocity is a cognitively demanding construct, we expect the cognitive measures to be high predictors for detecting the components of reciprocity, especially in older adults. Steward et al. [27] determined that detecting truth relied on attentional resources, while according to Sporer [28], deception is likely linked to working memory. Moreover, individuals with higher fluid intelligence abilities were shown to be more cooperative than people with lower fluid intelligence [18]. Therefore, we hypothesize that fluid intelligence will be a predictor of cooperation and working memory a predictor of deception.

For the personality dispositions, we expect the personality dimensions to lose relevance through age as cognitive functioning appears to play a unique role in vulnerability to social tricks in aging [19]. We hypothesize that a person with a higher degree of empathic concern will be better at detecting a situation of deception or cheating, since empathic concern is characterized by experiencing feelings of warmth and compassion for others experiencing negative situations, such as a feeling of protection for people that are taken advantage of. No direct link has been found in the literature.

## 2. Materials and Methods

### 2.1. Participants

The study involved 307 participants (age range: 20–96 years old) divided into three age groups: 98 younger adults (48 women, age range: 20–39, M = 27.43, SD = 5.51), 106 middle-aged adults (58 women, age range: 40–64, M = 53.14, SD = 6.71), and 103 community-dwelling older adults (53 women, age range: 65–96, M = 74.83, SD = 8.03). The participants were healthy community-based volunteers recruited in northern and central Italy through various methods, including word-of-mouth, flyers, e-mail, and social network messaging. Additionally, social service organizations (e.g., University of Third Age) were involved to reach middle-aged and older adults. All participants were required to fill out a demographic questionnaire to screen out individuals who had a history of substance abuse, psychiatric, or neurological disorders. For older adults, we used the Mini-Mental State Examination [29] as a cognitive screening tool to exclude participants with a score lower than 24. None of the participants were excluded based on the above criteria. A vocabulary test (drawn by the Primary Mental Abilities test) [30] was also included in the study as a control variable of crystallized intelligence. The study was conducted between December 2021 and April 2022, and it was approved by the Ethical Committee of the Department of Brain and Behavioral Sciences of the University of Pavia (n. 091/21). Additionally, informed consent was obtained from all participants.

### 2.2. Measures

#### 2.2.1. Cognitive Measures

##### Fluid Intelligence

We measured fluid intelligence through Raven’s Standard Progressive Matrices test (RPM) [31]. RPM is a non-verbal assessment of abstract reasoning, containing 48 items divided into four series (ranging from A to D) of 12 items each. Each item comprises a geometric design missing a piece. The participant must select the missing piece that would complete the geometric design out of six or eight given choices. Responses were scored 1 for a correct answer and 0 for an incorrect answer (possible range: 0–48).

##### Working Memory

The Backward Digit Span task (drawn by Wechsler Adult Intelligence Scales) [32] was used as a standardized measure of working memory. First, digit sequences extending from 2 to 8 digits are presented orally to the participants. They are then asked to repeat it in inverted order. The overall score comprises the total number of correctly recalled digits prior to failing two consecutive sequences at any one span size. Possible scores can range from 2 to 8.

#### 2.2.2. Reciprocity Detection and ToM Competences

The detection of cooperation and ToM abilities were assessed with a non-verbal task, the Mind Picture Story-Theory of Mind Questionnaire (MPS-TOMQ) of Calso et al. [11] based on the Theory of Mind Picture Story Task (TMPS) of Brüne [10]. We decided to use the version of Calso et al. [11] because they likely modified the pictures and stories to depict older characters and familiar situations to older adults. This task examines the ability to detect deception and cooperation in the context of deception (negative) and cooperation (positive), along with ToM performance assessing the ability to understand false and true beliefs. Thus, the task comprises six stories: two deception stories, two cooperation stories, and two mixed stories portraying both cooperation and deception, and each story, is formed by four pictures. 

As a first step, the experimenter placed the four pictures of each story in front of the participant in a mixed order. The participant is then asked to organize the pictures in a meaningful sequence to form the story. This part is the Mind Picture Story (MPS) and was evaluated through accuracy and reaction time. The participant is timed while placing the pictures together and a score is given to the sequence in which the pictures are placed. 

A score of two points is given for the correct sequence of the first and fourth pictures each and a score of one point for the third and fourth pictures, respectively if correctly placed. Thus, each story is scored on a range from 0 to 6. The total MPS score for the six stories ranges between 0 and 36.

Once the participant completes the logical sequencing of the pictures, the experimenter evaluates if the sequence is correct and if not, puts them in the exact order.

The second part of the task comprises a series of questions on the story evaluating the detection of cooperation and deception, and multiple aspects of ToM (first-, second-, and third-order beliefs and false beliefs). This part is called the Theory of Mind Questionnaire (TOMQ) and is originally divided into nine subscales, which we decided to maintain: reality, first-order true belief, first-order false belief, second-order true belief, second-order false belief, third-order false belief, cooperation, deception, and cheating. This questionnaire contains 24 items (to investigate reciprocity in positive and negative settings and to have the same number of reciprocity and deception items, we added a reciprocity question in one of the mixed stories), two reality items, two first-order true belief items, three first-order false belief items, two second-order true belief items, three second-order false belief items, two third-order false belief items, four cooperation items, four deception items, and two cheating items. The cooperation and deception items are not only in the cooperation and deception stories, respectively, but also in the mixed stories.

In contrast to the original scoring, we decided to score every item between 0 (the answer is not correct), 1 (partially correct), and 2 (completely correct). Each subscale was scored separately, and their range depends on the number of items.

These two parts are repeated for each story and the total score of the MPS-TOMQ is between 0 and 84.

#### 2.2.3. Personality Dispositions

We investigated three different types of personality dispositions: agreeableness, altruism, and empathic concern.

##### Big Five Questionnaire (BFQ)

We measured agreeableness through the 24 items of the Agreeableness scale of the Big Five questionnaire [33], which assesses the disposition to be cooperative, polite, kind, and friendly. The Italian version of the Big Five Agreeableness trait is divided into two dimensions, which are cordiality and cooperativeness. Items are rated on a scale ranging from 1 (very inaccurate) to 5 (very accurate).

##### Elderly Care Research Center (ECRC) Altruistic Scale

Altruism was estimated by the ECRC altruistic scale [34] to evaluate altruistic attitudes and orientations. The ECRC Altruistic scale comprises 5 items (e.g., “I enjoy doing things for others”, “I try to help others, even if they do not help me”) gauged on a 5-point Likert scale (namely, strongly disagree = 1, disagree = 2, neither agree nor disagree = 3, agree = 4, strongly agree = 5).

##### Interpersonal Reactivity Index (IRI)

To evaluate empathic concern, we used the Italian version of the empathic concern subscale from the IRI [35]. This subscale contains seven items quantified by a 5-point Likert scale ranging from 0 (Does not describe me well) to 5 (Describes me very well). This subscale determines the participants’ tendency to experience feelings of concern or compassion towards others.

#### 2.2.4. Procedure

Participants were tested individually during two sessions separated by a week. In the first session, after obtaining consent, participants first compiled a brief demographic questionnaire, and only participants over 65 years old underwent the MMSE to ensure eligibility for the study (MMSE > 24). Then, participants performed the vocabulary test and the Backward Digit Span. Subsequently, they completed the questionnaires in the following order: agreeableness scale of the BFQ, empathic concern subscale of the IRI, and the ECRC altruistic scale. In the second session, participants carried out the MPS-TOMQ and the Raven’s Progressive Matrices. The order of administration of the tests and questionnaires were the same for the three age groups.

### 2.3. Statistical Analysis

Intending to identify differences between the three age groups, we computed various one-way analyses of variances (ANOVAs) on background variables (i.e., years of education, vocabulary), cognitive measures (i.e., working memory and fluid intelligence), and personality dispositions (i.e., altruism, empathic concern, and the two dimensions of agreeableness: cordiality and cooperativeness). In order to test our first aim, we ran analyses of covariances (ANCOVAs) on the cooperation, deception, cheating, and ToM items from the MPS-TOMQ task to examine possible age group differences, controlling for years of education.

Next, to investigate our second aim, we used separate hierarchical multiple linear regression analyses to test which variables predicted the detection of cooperation, deception, and cheating in the three age groups. All analyses were performed using Rstudio version 4.1.2 [36].

## 3. Results

### 3.1. Preliminary Analysis on Background Variables

Table 1 holds the descriptive statistics of the background variables. The results of the ANOVA and post hoc analyses showed that there was a difference in level of education between the three age groups (*F*(2, 304) = 49.1, *p* < 0.001). Younger adults revealed a higher level of education than middle-aged adults and both younger and middle-aged adults had a higher level of education than older adults. There were no age-related differences for the vocabulary test (*F*(2, 304) = 2.136, *p* = 0.12). The descriptive statistic for the cognitive and personality variables can be found in the Appendix A.

### 3.2. Age Differences in the MPS-TOMQ 

Table 2 provides the descriptive statistics of all the reciprocity item scores of the MPS-TOMQ. One of the main aims of this study was to examine if there were age-related differences in the detection of cooperation, deception and cheating and in false belief understanding. As aforementioned, our sample demonstrated a significant age-related difference in years of education between all age groups. For this reason, various ANCOVAs controlling for years of education and post hoc analyses were conducted on all the items of the task. 

For the global score of detection of cooperation, no effect of age was revealed (*F*(2, 302) = 2.45, *p* = 0.088) as well as no effect of years of education (*F*(1, 302) = 0.23, *p* = 0.63), showing that younger, middle-aged, and older adults did not differ in their detection of cooperation. However, when the cooperation variable was subdivided into type of story, a significant age group difference emerged for cooperation in cooperation story (*F*(2, 302) = 3.93, *p* = 0.021) with no effect of years of education (*F*(1, 302) = 1.10, *p* = 0.30), while no differences among the age groups were found in cooperation in mixed stories (*F*(2, 302) = 0.49, *p* = 0.62) along with no effects of the level of education (*F*(1, 302) = 3.84, *p* = 0.051). Both younger and middle-aged adults displayed a greater detection of cooperation in cooperation stories than older adults, but no significant difference was found between younger and middle-aged adults. On the other hand, a main effect of age groups was present not only for deception detection (*F*(2, 302) = 4.86, *p* = 0.0084) globally, but also when subdivided by type of story (deceptive story: *F*(2, 302) = 3.51, *p* = 0.031; mixed story: *F*(2, 302) = 5.14, *p* = 0.0064), with no effect of years of education on all the items (deception: *F*(1, 302) = 0.78, *p* = 0.38; in deceptive story: *F*(1, 302) = 1.003, *p* = 0.32; in mixed story: *F*(1, 302) = 0.068, *p* = 0.79). The post hoc analysis divulged that younger and middle-aged adults exhibited a better deception detection compared to older adults. Once again, no significant differences emerged between younger and middle-aged adults. Instead, younger adults showed a significantly greater ability to detect deception in a deceptive story compared to older adults. Middle-aged adults were significantly more able to identify deception in mixed stories than older adults. Similar results were observed for the detection of cheating (*F*(2, 302) = 5.99, *p* = 0.04), with no effect of years of education (*F*(2, 302) = 3.43, *p* = 0.065). The ability for younger adults to detect cheating was significantly higher than for older adults. The descriptive values and analysis for the ToM items are reported in the Appendix A.

### 3.3. Regression Analysis in the Different Age Groups

To examine potential predictor effects of personality dispositions, cognitive measures, and true and false belief items on reciprocity, deception, and cheating detection in younger, middle-aged, and older adults, we applied hierarchical regression analyses with three stages (detailed in Table 3, Table 4 and Table 5). The first step only included personality dispositions, namely, the two dimensions of agreeableness, i.e., cordiality and cooperativeness, empathic concern, and altruism as predictors. The second step added the cognitive measures of working memory and fluid intelligence to the predictors. In the third step, true belief items were added as predictors for the cooperation items and false belief items were added as predictors to the deception detection items as suggested by the literature [14]. 

In the younger adults’ sample, the first step of the model was never significant for none of the outcome variables. Adding the cognitive measures, the model was significant and increased *R^2^* significantly in cooperation in mixed story (*F*(6, 91) = 3.592, *p* = 0.003), deception (*F*(6, 91) = 3.747, *p* = 0.002), deception in mixed story (*F*(6, 91) = 2.207, *p* = 0.049), and cheating (*F*(6, 91) = 2.214, *p* = 0.049). Working memory was a positive and significant predictor for all these variables. The ToM items of belief in the third step added significance to the model and a significant variation of *R^2^* in a significant model for deception (*F*(9, 88) = 6.571, *p* < 0.001), deception in deception story (*F*(9, 88) = 6.618, *p* < 0.001), and cheating (*F*(9, 88) = 3.279, *p* = 0.002). However, the *R^2^* variation was not significant for cooperation. Second- and third-order false beliefs were predictors of deception in deception story, and only third-order false belief remained a positive predictor of deception. Whereas first- and third-order false beliefs significantly predicted the detection of cheating, negatively and positively, respectively. In this third step, working memory remained a significant predictor for deception. 

The regression models in the middle-aged group showed that the first step of the model concerning the personality dispositions was not significant for any of the outcomes. The addition of the cognitive measure in the second step varied *R^2^* significantly in cooperation (*F*(6, 99) = 2.325, *p* = 0.038), deception (*F*(6, 99) = 9.894, *p* < 0.001), deception in deception story (*F*(6, 99) = 3.324, *p* = 0.005), deception in mixed story (*F*(6, 99) = 7.127, *p* < 0.001), and cheating (*F*(6, 99) = 2.986, *p* = 0.01). Working memory was a significant predictor for all variables of reciprocity and cooperativeness was a significant predictor of deception in mixed story. In the third step, adding the ToM items was significant and increased *R^2^* for cooperation (*F*(9, 96) = 2.872, *p* = 0.007), deception (*F*(9, 96) = 13.21, *p* < 0.001), deception in deception story (F(9,96) = 7.134, *p* < 0.001), and cheating (*F*(9, 96) = 5.28, *p* < 0.001). Second-order true belief was a predictor of cooperation, whereas third-order false belief was a predictor of the global score of deception, deception in deception story, and cheating.

For older adults, the first step of the hierarchical regression revealed that once again none of the models were significant. Adding the cognitive measure in the second step generated strong predictors for the model. The analysis uncovered that reasoning was a significant predictor of cooperation and cooperation in mixed story, while working memory was a significant predictor of the global score of deception and deception in deception story in older adults. Both cognitive measures significantly predicted deception in mixed story. In the third step, adding the ToM measures to the model determined that true beliefs were a predictor of cooperation and false beliefs a predictor of deception. Therefore, older adults with a higher level of true beliefs of first- and second-order reasoning were better at detecting cooperation, while a higher level of false beliefs of second- and third-order reasoning in older adults leads to a higher level of deception and cheating detection. Table 6 reports a schematic synthesis of the predictors of the different reciprocity dimensions in the three age groups.

## 4. Discussion

The present study first aimed at investigating age-related differences in the detection of the different components of reciprocity: deception, cheating, and cooperation. With this aim in mind, we used a task assessing various reciprocity items through stories with a cooperative, deceptive, and mix of both contexts including Theory of Mind (ToM) items. Second, it intended to examine the predictors of reciprocity, such as cognitive measures (namely, fluid intelligence and working memory), personality dispositions (namely, the cooperativeness and cordiality dimensions of agreeableness, altruism, and empathic concern), and true and false beliefs. 

Regarding the first aim, the results showed that younger and middle-aged adults were better at detecting cooperation, especially in cooperation stories, and deception in both deception and mixed stories, than older adults controlling for the years of education, which is in line with our expectations. Furthermore, younger adults had a significant higher ability to detect cheating compared to older adults. These results indicate that the decline in the detection of the reciprocity components starts above 65 years old. These findings are consistent with previous results of Calso et al. [11,12] as well as Raimo et al. [13] on reciprocity. However, it is important to note that Calso et al. [11,12] only considered scores depending on the type of story; therefore, making no distinction between the reciprocity items and ToM items, as well as a total score of the task, whereas Raimo et al. [13] only analyzed the total score, regrouping all the subcomponents.

Hence, the novelty of these results lies in the analysis of the cooperation and deception subscales by type of story, as a pure measure of reciprocity. De facto, we found different age patterns. Younger adults were significantly better at detecting deception in deception stories, whereas middle-aged adults were better at detecting deception in mixed stories compared to older adults. Both younger and middle-aged adults were better at detecting cooperation in cooperation stories in comparison to older adults. We expected to find different age-related differences for the various types of stories, but surprisingly and contradictory to what we hypothesized, age differences in the mixed stories were only found in deception and not in cooperation. This may be interpreted by the difficulty of the task. Indeed, we found that performance scores in deception were lower compared to cooperation. This could be attributed to the truth and positivity bias commonly observed in older adults, coupled with their reduced ability to discern lies [37]. Consequently, older adults may exhibit a slightly worse proficiency in detecting false beliefs compared to true beliefs, leading to increased difficulty in detecting deception.

Concerning the second aim, reciprocity, deception, and cheating detection, appeared to be predicted by true and false beliefs, and the cognitive measures of working memory and fluid intelligence, depending on the age groups, as expected. Although, the personality dispositions of agreeableness, empathic concern, and altruism, did not predict any component of reciprocity for none of the age groups.

Consistent with our expectations, ToM emerged as a robust predictor of the reciprocity dimensions, particularly in older adults. For younger adults, only false beliefs demonstrate a significant predictive role in deception and cheating. Conversely, the understanding of true and false beliefs proved to be a dominant predictor in middle-aged and older adults for the reciprocity components. True beliefs revealed to be predictive of cooperation, while false beliefs were predictive of deception and cheating. These results are in line with the findings of Lissek et al. [14] on the association between belief reasoning and the detection of cooperation and deception. Interestingly, the prediction strengthened with age, as in the older adult group, both first- and second-order true beliefs significantly predicted cooperation. Moreover, not only third-order false belief, but also second-order false belief predicted the global score of deception, deception in deception story, and cheating. These different associations also confirm that the two constructs, reciprocity and ToM, can be investigated separately in the present task and that MPS-TOMQ represents a useful measure of reciprocity.

As hypothesized cognitive measures were found to predict the detection of reciprocity, a more pronounced effect was observed for older adults. Specifically, in younger adults, working memory was predictive of their capacity to detect deception but not cooperation and cheating. Moreover, in the middle-aged group, only working memory occurred as a cognitive predictor of the different reciprocity dimensions, e.g., cooperation, deception, and cheating. In contrast, older adults exhibited the use of distinct cognitive abilities for different reciprocity dimensions. For this group, working memory revealed to be a strong predictor for the detection of deception, while reasoning was a predictor for the detection of cooperation. However, the distinctive use of cognitive resources between cooperation and detection as found in the literature [14,27,28] was only observed in older adults, against our expectations.

Contrary to our hypothesis, personality dispositions did not play a predictive role in the younger adult group. However, empathic concern in younger adults initially showed significance as a predictor of cooperation, along with working memory, before the inclusion of ToM measures in the regression analysis. Nevertheless, across all age groups, personality dispositions did not predict cooperation, deception, and cheating once cognitive or socio-cognitive factors were added to the regression. Overall, reciprocity dimensions are mainly predicted by cognitive abilities, and ToM competences. However, the reciprocity dimensions are explained by different variables across age groups. 

A limit of this study is the observational stance of the task we used. It would possibly be more appropriate to evaluate reciprocity and its dimensions adding to the present task investigation by using more ecological and interactive tasks, such as the interactive drawing task [38], already used to evaluate the reciprocal behavior of children with autism spectrum disorder. Moreover, future research might explore whether providing training in ToM abilities in older adults could potentially have a transfer effect on their ability to detect reciprocity. Another limitation of our study could be the fact that the reciprocity detection task was solely visual, while older adults have shown more difficulties in detecting visual modalities of deception rather than audio and audiovisual modalities [6]. In the future, it could be very useful to add other modalities to verify if the decline in reciprocity could be related to the modality or is indeed a decline of the reciprocity dimensions. Additionally, for this study we recruited a convenience sample which is not necessarily representative of the entire population. Finally, the cross-sectional estimates of age-related differences in the present study might be biased by cohort effects, such as sociocultural differences in years of education. However, we controlled all analyses for age differences in education level on reciprocity and its components, and we did not detect any effects of education.

To conclude, using the different subscales and types of stories of the MPS-TOMQ allowed us to make a clear distinction between cooperation, deception, and cheating demonstrating the different predictive roles of cognition and ToM abilities in a precise context and age group. It could, therefore, be interesting for future studies to maintain these distinctions in coding, and further analyze the source of the difference in difficulty between cooperation in a cooperative or mixed story, as well as the divergence between deception and cooperation in a similar story. Exploring the influence of the story on the various components of reciprocity is indeed an intriguing avenue for further investigation. Moreover, this study uncovered that the decline in the detection of reciprocity dimensions occurs in older age, specifically starting from 65 years old, by providing a continuum of three age categories. Importantly, this specific regression analysis, acknowledging the ties between beliefs and reciprocity, supports Calso’s approach of using a ToM task to evaluate deception and confirms that cognition and social cognition are important concepts to detect reciprocity. Further studies should continue investigating this topic, regrouping the reciprocity dimensions through various contexts using a naturalistic task.

## Figures and Tables

**Table 1 behavsci-13-01007-t001:** Background variables descriptive statistics by age group.

	Younger Adults (*n* = 98)		Middle-Aged Adults (*n* = 106)		Older Adults (*n* = 103)	
	Mean (SD)	Range	Mean (SD)	Range	Mean (SD)	Range
Age	27.43 (5.51)	20–39	53.14 (6.71)	40–64	74.83 (8.03)	65–96
Gender (F/M)	48/50		58/48		53/50	
Education	15.48 (2.66)	8–20	13.30 ^(^***^)^ (3.40)	8–24	10.5 ^(^***^,+++)^ (4.42)	3–18
Vocabulary	40.12 (6.40)	18–49	40.97 (8.40)	10–50	38.63 (9.58)	9–50

Note: A significant difference from the young adult group is represented by (***) *p* < 0.001; a significant difference from the middle-aged group is represented by (+++) *p* < 0.001.

**Table 2 behavsci-13-01007-t002:** Adjusted descriptive statistics for the MPS-TOMQ task divided by age group.

	Younger Adults		Middle-Aged Adults		Older Adults	
	Mean (SE)	Range	Mean (SE)	Range	Mean (SE)	Range
Cooperation	6.77 (0.15)	4–8	6.84 (0.14)	4–8	6.40 (0.15)	1–8
Cooperation story	3.38 (0.09)	1–4	3.36 (0.09)	1–4	3.03 ^(^*^,+)^ (0.09)	0–4
Mixed story	3.39 (0.09)	1–4	3.48 (0.08)	1–4	3.37 (0.09)	0–4
Deception	4.73 (0.17)	2–7	4.69 (0.15)	2–7	4.05 ^(^*^,+)^ (0.17)	0–7
Deception story	2.25 (0.13)	0–4	2.14 (0.12)	0–4	1.76 ^(^*^)^ (0.13)	0–4
Mixed story	3.19 (0.06)	2–4	3.24 (0.06)	1–4	2.97 ^(++)^ (0.06)	0–4
Cheating	3.17 (0.10)	1–4	3.07 (0.09)	0–4	2.79 ^(^*^)^ (0.10)	0–4

Note: significant difference from the young adult group is represented by (*) *p* < 0.05; a significant difference from the middle-aged group is represented by (++) *p* < 0.01, (+) *p* < 0.05.

**Table 3 behavsci-13-01007-t003:** Hierarchical linear regression analysis in the young adult group.

	Cooperation			Cooperation in Cooperation Story			Cooperation in Mixed Story					
Predictors	B	SE B	ß	B	SE B	ß	B	SE B	ß			
**Step 1**												
Agreeableness												
Cordiality	0.022	0.024	0.118	0.008	0.017	0.058	0.015	0.014	0.136			
Cooperativeness	−0.051	0.037	−0.217	−0.013	0.026	−0.079	−0.038	0.021	−0.282			
Empathic concern	0.084	0.046	0.265	0.047	0.032	0.216	0.036	0.026	0.201			
Altruism	−0.004	0.065	−0.011	−0.014	0.045	−0.049	0.009	0.037	0.041			
**Step 2**												
Working memory	0.226	0.097	0.235 *	0.034	0.071	0.051	0.192	0.053	0.158 ***			
Fluid intelligence	0.040	0.036	0.117	0.009	0.026	0.038	0.031	0.020	0.346			
Agreeableness												
Cordiality	0.010	0.024	0.054	0.005	0.018	0.040	0.005	0.014	0.046			
Cooperativeness	−0.049	0.036	−0.209	−0.013	0.026	−0.079	−0.036	0.020	−0.268			
Empathic concern	0.107	0.046	0.340 *	0.051	0.033	0.234	0.056	0.025	0.308			
Altruism	−0.015	0.064	−0.037	−0.015	0.046	−0.053	0.00004	0.035	0.0002			
**Step 3**												
Working memory	0.185	0.097	0.193	0.003	0.071	0.004	0.183	0.055	0.330			
Fluid intelligence	0.066	0.037	0.191	0.028	0.027	0.116	0.038	0.021	0.192			
Agreeableness												
Cordiality	0.016	0.024	0.086	0.009	0.018	0.071	0.007	0.014	0.062			
Cooperativeness	−0.051	0.036	−0.217	−0.013	0.026	−0.080	−0.038	0.020	−0.281			
Empathic concern	0.117	0.045	0.373	0.058	0.033	0.266	0.059	0.025	0.327			
Altruism	−0.027	0.063	−0.067	−0.023	0.046	−0.081	−0.004	0.035	−0.018			
True belief first order	0.193	0.139	0.140	0.183	0.101	0.191	0.010	0.078	0.012			
True belief second order	0.379	0.201	0.194	0.244	0.146	0.180	0.135	0.113	0.121			
*R^2^*	*R^2^* = 0.013 for step 1 *∂R^2^* = 0.068 * for step 2 *∂R^2^* = 0.047 for step 3			*R^2^* = −0.012 for step 1 *∂R^2^* = 0.004 for step 2 *∂R^2^* = −0.003 for step 3			*R^2^* = −0.007 for step 1 *∂R^2^* = 0.0007 *** for step 2 *∂R^2^* = −0.013 for step 3					
	**Deception**			**Deception in Deception Story**			**Deception in Mixed Story**			**Cheating**		
**Predictors**	**B**	**SE B**	**ß**	**B**	**SE B**	**ß**	**B**	**SE B**	**ß**	**B**	**SE B**	**ß**
**Step 1**												
Agreeableness												
Cordiality	−0.039	0.026	−0.189	−0.021	0.023	−0.120	0.002	0.010	0.027	−0.021	0.016	−0.170
Cooperativeness	−0.033	0.040	−0.130	−0.022	0.035	−0.100	−0.008	0.014	−0.086	−0.023	0.024	−0.147
Empathic concern	0.075	0.050	−0.219	0.088	0.043	0.296	−0.014	0.018	−0.119	0.033	0.030	0.157
Altruism	0.008	0.070	0.018	−0.046	0.061	−0.121	0.005	0.025	0.030	0.048	0.043	0.179
**Step 2**												
Working memory	0.393	0.101	0.374 ***	0.233	0.091	0.257 *	−0.051	0.039	−0.137	0.162	0.064	0.253 *
Fluid intelligence	−0.029	0.037	−0.078	0.020	0.034	0.062	−0.006	0.014	−0.044	−0.022	0.024	−0.098
Agreeableness												
Cordiality	−0.044	0.025	−0.216	−0.030	0.023	−0.171	0.004	0.010	0.057	−0.022	0.016	−0.175
Cooperativeness	−0.019	0.038	−0.072	−0.018	0.034	−0.060	−0.009	0.015	−0.095	−0.016	0.024	−0.100
Empathic concern	0.101	0.048	0.294 *	0.109	0.043	0.367 *	−0.019	0.018	−0.158	0.042	0.030	0.201
Altruism	−0.026	0.066	−0.059	−0.060	0.060	−0.158	0.007	0.026	0.048	0.032	0.042	0.121
**Step 3**												
Working memory	0.231	0.098	0.220 *	0.076	0.084	0.084	−0.029	0.043	−0.079	0.084	0.066	0.132
Fluid intelligence	−0.057	0.034	−0.153	−0.010	0.029	−0.032	−0.007	0.015	−0.049	−0.027	0.023	−0.119
Agreeableness												
Cordiality	−0.031	0.023	−0.150	−0.016	0.019	−0.093	0.003	0.010	0.043	−0.018	0.015	−0.145
Cooperativeness	−0.017	0.033	−0.064	−0.016	0.029	−0.070	−0.008	0.015	−0.086	0.014	0.023	−0.092
Empathic concern	0.047	0.043	0.135	0.054	0.037	0.180	−0.014	0.019	−0.118	0.029	0.030	0.137
Altruism	0.021	0.059	0.048	−0.012	0.051	−0.032	0.003	0.026	0.021	0.043	0.040	0.163
False belief first order	−0.011	0.113	−0.009	0.020	0.097	0.018	0.008	0.050	0.017	−0.175	0.077	−0.228 *
False belief second order	0.226	0.133	0.182	0.237	0.114	0.222 *	0.034	0.059	0.078	0.120	0.090	0.160
False belief third order	0.441	0.138	0.364 **	0.424	0.119	0.404 ***	−0.073	0.061	−0.171	0.191	0.094	0.259 *
*R^2^*	*R^2^* = 0.02 for step 1 *∂R^2^* = 0.14 *** for step 2 *∂R^2^* = 0.20 *** for step 3			*R^2^* = 0.02 for step 1 *∂R^2^* = 0.07 ** for step 2 *∂R^2^* = 0.28 *** for step 3			*R^2^* = −0.02 for step 1 *∂R^2^* = 0.02 for step 2 *∂R^2^* = 0.02 for step 3			*R^2^* = 0.02 for step 1 *∂R^2^* = 0.14 * for step 2 *∂R^2^* = 0.20 ** for step 3		

Note: *** *p* < 0.001, ** *p* < 0.01, * *p* < 0.05

**Table 4 behavsci-13-01007-t004:** Hierarchical linear regression analysis for the middle-aged adult group.

	Cooperation			Cooperation in Cooperation Story			Cooperation in Mixed Story					
Predictors	B	SE B	ß	B	SE B	ß	B	SE B	ß			
**Step 1**												
Agreeableness												
Cordiality	0.022	0.024	0.105	0.004	0.017	0.030	0.018	0.014	0.153			
Cooperativeness	−0.013	0.029	−0.060	−0.004	0.020	−0.029	−0.009	0.016	−0.072			
Empathic concern	−0.039	0.033	−0.142	−0.020	0.023	−0.108	−0.19	0.019	−0.122			
Altruism	−0.036	0.050	0.090	0.021	0.034	0.078	0.015	0.028	0.068			
**Step 2**												
Working memory	0.332	0.099	0.334 **	0.194	0.068	0.289	0.137	0.056	0.248			
Fluid intelligence	−0.016	0.023	−0.068	−0.029	0.016	−0.179	0.013	0.013	0.096			
Agreeableness												
Cordiality	0.014	0.024	0.070	0.003	0.016	0.020	0.012	0.013	0.101			
Cooperativeness	0.0003	0.028	0.002	0.005	0.019	0.032	−0.005	0.016	−0.037			
Empathic concern	−0.019	0.033	−0.068	−0.011	0.022	−0.056	−0.008	0.019	−0.54			
Altruism	0.029	0.048	0.073	0.016	0.033	0.061	0.013	0.027	0.058			
**Step 3**												
Working memory	0.264	0.108	0.266 *	0.139	0.074	0.207	0.125	0.062	0.227			
Fluid intelligence	−0.012	0.023	−0.049	−0.025	0.016	−0.152	0.013	0.014	0.096			
Agreeableness												
Cordiality	0.022	0.023	0.104	0.007	0.016	0051	0.015	0.013	0.126			
Cooperativeness	0.007	0.028	0.032	−0.010	0.019	0.066	−0.003	0.016	−0.024			
Empathic concern	−0.032	0.032	−0.114	−0.019	0.022	−0.102	−0.013	0.019	−0.081			
Altruism	0.022	0.047	0.056	0.011	0.032	0.040	0.011	0.027	0.052			
True belief first order	0.011	0.129	0.009	0.046	0.089	0.057	−0.035	0.074	−0.053			
True belief second order	0.609	0.226	0.270 **	0.364	0.156	0.238	0.245	0.130	0.195			
*R^2^*	*R^2^* = −0.01 for step 1 *∂R^2^* = 0.10 ** for step 2 *∂R^2^* = 0.08 * for step 3			*R^2^* = −0.03 for step 1 *∂R^2^* = 0.09 for step 2 *∂R^2^* = −0.0004 for step 3			*R^2^* = −0.009 for step 1 *∂R^2^* = 0.07 for step 2 *∂R^2^* = 0.03 for step 3					
	**Deception**			**Deception in Deception Story**			**Deception in Mixed Story**			**Cheating**		
**Predictors**	**B**	**SE B**	**ß**	**B**	**SE B**	**ß**	**B**	**SE B**	**ß**	**B**	**SE B**	**ß**
**Step 1**												
Agreeableness												
Cordiality	0.018	0.029	0.070	−0.003	0.022	−0.014	0.007	0.010	0.081	−0.025	0.018	−0.167
Cooperativeness	−0.067	0.034	−0.248	−0.011	0.026	−0.054	−0.036	0.012	−0.370	0.011	0.021	0.064
Empathic concern	−0.042	0.040	−0.126	−0.019	0.030	−0.078	−0.005	0.014	−0.042	−0.023	0.025	−0.113
Altruism	0.070	0.060	0.145	0.035	0.045	0.099	0.008	0.021	0.043	0.029	0.037	0.100
**Step 2**												
Working memory	0.707	0.101	0.590 ***	0.323	0.086	0.368 ***	0.187	0.039	0.430 **	0.273	0.072	0.374 ***
Fluid intelligence	−0.006	0.024	−0.021	0.030	0.020	0.139	−0.025	0.009	−0.236 **	−0.006	0.017	−0.032
Agreeableness												
Cordiality	−0.002	0.024	−0.010	−0.017	0.020	−0.092	0.006	0.009	0.061	−0.033	0.017	−0.022
Cooperativeness	−0.040	0.029	−0.147	−0.0001	0.024	−0.0007	−0.028	0.011	−0.281	0.021	0.020	0.130
Empathic concern	0.005	0.033	0.014	0.006	0.028	0.024	0.005	0.013	0.039	−0.005	0.024	−0.026
Altruism	0.056	0.049	0.117	0.029	0.042	0.084	0.003	0.019	0.019	0.024	0.035	0.082
**Step 3**												
Working memory	0.518	0.093	0.432 ***	0.169	0.079	0.192 *	0.184	0.041	0.423	0.166	0.069	0.228 *
Fluid intelligence	−0.020	0.021	−0.070	0.013	0.018	0.060	−0.022	0.009	−0.209	−0.010	0.016	−0.055
Agreeableness												
Cordiality	0.030	0.022	0.118	0.016	0.019	0.088	0.002	0.010	0.023	−0.019	0.016	−0.125
Cooperativeness	−0.051	0.026	−0.189	−0.019	0.022	−0.096	−0.022	0.011	−0.023	0.022	0.019	0.136
Empathic concern	0.018	0.029	0.052	0.017	0.025	0.069	0.005	0.013	0.037	0.0008	0.022	0.004
Altruism	0.012	0.043	0.025	−0.005	0.037	−0.014	0.002	0.019	0.009	−0.004	0.032	−0.014
False belief first order	−0.045	0.100	−0.033	0.011	0.085	0.108	−0.083	0.045	−0.169	−0.130	0.074	−0.158
False belief second order	0.164	0.100	0.127	0.093	0.085	0.099	0.023	0.045	0.049	0.092	0.074	0.118
False belief third order	0.461	0.090	0.405 ***	0.395	0.077	0.474 ***	−0.003	0.040	−0.007	0.267	0.067	0.386 ***
*R^2^*	*R^2^* = 0.02 for step 1 *∂R^2^* = 0.31 *** for step 2 *∂R^2^* = 0.18 *** for step 3			*R^2^* = −0.03 for step 1 *∂R^2^* = 0.158 *** for step 2 *∂R^2^* = 0.23 *** for step 3			*R^2^* = 0.08 ** for step 1 *∂R^2^* = 0.19 *** for step 2 *∂R^2^* = 0.02 for step 3			*R^2^* = −0.01 for step 1 *∂R^2^* = 0.31 *** for step 2 *∂R^2^* = 0.18 *** for step 3		

Note: *** *p* < 0.001, ** *p* < 0.01, * *p* < 0.05

**Table 5 behavsci-13-01007-t005:** Hierarchical linear regression analysis in the older adult group.

	Cooperation			Cooperation in Cooperation Story			Cooperation in Mixed Story					
Predictors	B	SE B	ß	B	SE B	ß	B	SE B	ß			
**Step 1**												
Agreeableness												
Cordiality	0.056	0.033	0.220	0.033	0.020	0.220 ^+^	0.023	0.020	0.142			
Cooperativeness	0.037	0.037	0.128	−0.011	0.022	−0.065	0.048	0.023	0.265 *			
Empathic concern	−0.009	0.043	−0.021	−0.013	0.026	−0.053	0.004	0.027	0.018			
Altruism	−0.027	0.064	−0.053	0.044	0.038	0.147	−0.071	0.040	−0.223			
**Step 2**												
Working memory	0.135	0.125	0.118	0.081	0.079	0.046	0.055	0.074	0.076			
Fluid intelligence	0.053	0.019	0.305 **	0.005	0.011	0.189	0.048	0.011	0.442 ***			
Agreeableness												
Cordiality	0.043	0.031	0.167	0.029	0.020	0.189	0.014	0.018	0.086			
Cooperativeness	0.009	0.036	0.031	−0.014	0.023	−0.081	0.023	0.021	0.125			
Empathic concern	0.015	0.043	0.036	−0.003	0.027	−0.013	0.018	0.025	0.070			
Altruism	−0.003	0.061	−0.005	0.049	0.039	0.164	−0.052	0.036	−0.164			
**Step 3**												
Working memory	0.018	0.116	0.015	0.016	0.012	0.003	0.002	0.071	0.003			
Fluid intelligence	0.040	0.017	0.231 *	−0.0003	0.078	0.372 ***	0.041	0.010	0.372 ***			
Agreeableness												
Cordiality	0.021	0.028	0.083	0.018	0.019	0.019	0.003	0.017	0.019			
Cooperativeness	0.004	0.032	0.017	−0.015	0.022	0.109	0.020	0.020	0.109			
Empathic concern	0.018	0.038	0.045	−0.001	0.025	0.076	0.019	0.023	0.076			
Altruism	−0.0001	0.054	−0.0002	0.051	0.036	−0.162	−0.051	0.033	−0.162			
True belief first order	0.392	0.152	0.236 *	0.230	0.102	0.155	0.162	0.093	0.155			
True belief second order	0.625	0.168	0.340 ***	0.262	0.113	0.314 ***	0.363	0.102	0.314 ***			
*R^2^*	*R^2^* = 0.04 for step 1 *∂R^2^* = 0.12 *** for step 2 *∂R^2^* = 0.19 *** for step 3			*R^2^* = 0.03 for step 1 *∂R^2^* = 0.21 *** for step 2 *∂R^2^* = −0.13 *** for step 3			*R^2^* = 0.03 for step 1 *∂R^2^* = 0.21 *** for step 2 *∂R^2^* = −0.13 *** for step 3					
	**Deception**			**Deception in Deception Story**			**Deception in Mixed Story**			**Cheating**		
**Predictors**	**B**	**SE B**	**ß**	**B**	**SE B**	**ß**	**B**	**SE B**	**ß**	**B**	**SE B**	**ß**
**Step 1**												
Agreeableness												
Cordiality	0.058	0.036	0.205	0.025	0.026	0.044	0.003	0.015	0.029	0.018	0.023	0.104
Cooperativeness	−0.031	0.041	−0.096	−0.013	0.030	−0.078	0.011	0.017	0.090	0.037	0.026	0.187
Empathic concern	−0.107	0.048	−0.242 *	−0.033	0.035	0.003	−0.028	0.019	−0.163	−0.016	0.030	−0.057
Altruism	0.058	0.071	0.104	0.068	0.052	0.211	0.002	0.029	0.007	−0.038	0.045	−0.111
**Step 2**												
Working memory	0.672	0.125	0.532 ***	0.301	0.102	0.332 **	0.135	0.008	0.273 *	0.164	0.084	0.210
Fluid intelligence	0.012	0.019	0.062	0.007	0.015	0.052	0.016	0.055	0.213	0.038	0.013	0.319 **
Agreeableness												
Cordiality	0.022	0.031	0.078	0.009	0.025	0.044	−0.006	0.014	−0.052	0.005	0.021	0.028
Cooperativeness	−0.038	0.036	−0.121	−0.018	0.030	−0.078	0.003	0.016	0.021	0.017	0.024	0.085
Empathic concern	−0.032	0.042	−0.071	0.001	0.034	0.003	−0.011	0.019	−0.063	0.008	0.029	0.030
Altruism	0.092	0.061	0.165	0.084	0.049	0.211	0.013	0.027	0.060	−0.018	0.041	−0.052
**Step 3**												
Working memory	0.540	0.110	0.427 ***	0.238	0.092	0.263	0.096	0.055	0.194	0.099	0.081	0.126
Fluid intelligence	−0.021	0.018	−0.107	−0.026	0.015	−0.189	0.012	0.009	0.163	0.019	0.013	0.158
Agreeableness												
Cordiality	0.030	0.027	0.104	0.019	0.022	0.092	−0.004	0.013	−0.034	0.006	0.020	0.031
Cooperativeness	−0.040	0.031	−0.126	−0.025	0.026	−0.109	0.004	0.015	0.036	0.014	0.023	0.070
Empathic concern	−0.021	0.037	−0.047	−0.001	0.031	−0.004	−0.008	0.018	−0.045	0.018	0.027	0.066
Altruism	0.005	0.053	0.010	0.027	0.044	0.068	−0.008	0.027	−0.036	−0.063	0.039	−0.182
False belief first order	−0.059	0.120	−0.040	0.105	0.100	0.098	−0.028	0.060	−0.048	−0.083	0.088	−0.089
False belief second order	0.238	0.092	0.245 *	0.210	0.077	0.301 **	0.003	0.046	0.009	0.216	0.068	0.358 **
False belief third order	0.423	0.095	0.368 ***	0.275	0.080	0.333 ***	0.147	0.048	0.325 **	0.120	0.070	0.168
*R^2^*	*R^2^* = 0.03 for step 1 *∂R^2^* = 0.28 *** for step 2 *∂R^2^* = 0.20 *** for step 3			*R^2^* = 0.003 for step 1 *∂R^2^* = 0.113 *** for step 2 *∂R^2^* = 0.23 *** for step 3			*R^2^* = −0.01 for step 1 *∂R^2^* = 0.15 *** for step 2 *∂R^2^* = 0.09 * for step 3			*R^2^* = 0.007 for step 1 *∂R^2^* = 0.28 *** for step 2 *∂R^2^* = 0.20 *** for step 3		

Note: *** *p* < 0.001, ** *p*< 0.01, * *p* < 0.05.

**Table 6 behavsci-13-01007-t006:** Summary of the significant predictors of the different reciprocity dimensions for each age group.

	Cooperation	Deception	Cheating
**Young adults**		Working memory Third-order false belief	First-order false belief Third-order false belief
**Middle-aged adults**	Working memory Second-order true belief	Working memory Third-order false belief	Working memory Third-order false belief
**Older adults**	Reasoning First-order true belief Second-order true belief	Working memory Second-order false belief Third-order false belief	Second-order false belief Third-order false belief

## Data Availability

The raw data supporting the conclusions of this article will be made available by the authors upon request, without undue reservation.

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
