# Peer review of "Cognitive Functions, Theory of Mind Abilities, and Personality Dispositions as Potential Predictors of the Detection of Reciprocity in Deceptive and Cooperative Contexts through Different Age Groups"

_behavsci, 2023, doi:10.3390/bs13121007_

Round 1

Reviewer 1 Report

Comments and Suggestions for Authors

Introduction

1.      The introduction provides a brief background on reciprocity and notes gaps in the literature regarding age differences and predictors of reciprocity detection. However, the specific research questions and hypotheses are not clearly stated. The authors should explicitly state what they aimed to examine.

2.      The rationale for using the MPS-TOMQ task should be explained more clearly. Why is this an appropriate measure for assessing cooperation, deception, cheating, and theory of mind?

3.       A brief outline of the study design, measures, and analysis approach should be provided at the end of the introduction.

Literature Review

1.      The literature review covers relevant concepts but lacks synthesis of prior empirical findings. The authors should expand on previous studies on age differences in reciprocity, deception, cooperation, and factors that may impact performance.

2.      There is the repetition of definitions already stated in the introduction. The literature review should focus more on summarizing and critiquing related research.

3.      More justification is needed for the hypothesized relationships between reciprocity detection, cognitive abilities, personality traits, and theory of mind. The existing literature should be used to build this rationale.

Methods

1.      Details on participant recruitment, selection criteria, and characteristics should be expanded. What was the full age range? How was eligibility determined for older adults?

2.      The measures are explained clearly, but more information on procedures and administration order is needed.

3.      The data analysis approach should be described briefly. What statistical tests were used to address each research question?

Results and Discussion

1.      The authors should relate the findings back to previous work and note consistencies or contradictions. How do these age differences compare to prior studies?

2.      The theoretical and practical implications should be expanded on. For example, how might these results inform interventions to improve reciprocity detection in older adults?

3.      Imitations of the study and future research directions should be discussed. Are there ways to improve the reciprocity measurement approach?

Reviewer 2 Report

Comments and Suggestions for Authors

This is an interested study of the theory of mind ability of adults in different age groups, and links with cognitive ability. My main concern here is the description of the statistical significance of findings - I appreciate there are degree of significance, but findings which are clearly not significant should not be described as being 'marginal' or 'approaching significance' as this cannot be known - all that can be stated is that in this study the finding are not significant. Therefore the results and discussion should be amended to clarify this, otherwise it becomes difficult to work out which results are significant. For detection of 'cheating' my reading of the results is that this was only significantly different between older adults and young adults, with no difference between middle aged and older adults - which isn't how the discussion reads. It's not clear if differences between middle aged and younger adults were explored or not? This would be of interest. 

Comments on the Quality of English Language

A few minor typos only. 

Reviewer 3 Report

Comments and Suggestions for Authors

Overall, I found the study very interesting, particularly because it addresses the social dimension in the older adult population, an area that has been insufficiently explored despite the increasing proportion of elderly individuals in the global population.
